# Genetic Resistance of Bovines to Theileriosis

**DOI:** 10.3390/ani12212903

**Published:** 2022-10-23

**Authors:** Diana Valente, Jacinto Gomes, Ana Cláudia Coelho, Inês Carolino

**Affiliations:** 1CIVG—Vasco da Gama Research Center, EUVG—Vasco da Gama University School, 3020-210 Coimbra, Portugal; 2Escola de Ciências Agrárias e Veterinárias, Universidade de Trás-os-Montes e Alto Douro, 5000-801 Vila Real, Portugal; 3Escola Superior Agrária de Elvas, Instituto Politécnico de Portalegre, 7350-092 Elvas, Portugal; 4CIISA—Centre for Interdisciplinary Research in Animal Health, Faculty of Veterinary Medicine, University of Lisbon, 1300-477 Lisbon, Portugal; 5Associate Laboratory for Animal and Veterinary Sciences (AL4AnimalS), Faculty of Veterinary Medicine, University of Lisbon, 1300-477 Lisbon, Portugal; 6Polo de Inovação da Fonte Boa—Estação Zootécnica Nacional, Instituto Nacional de Investigação Agrária e Veterinária, 2005-424 Santarém, Portugal; 7ISA—Instituto Superior de Agronomia, Universidade de Lisboa, 1349-017 Lisboa, Portugal

**Keywords:** tick-borne diseases, bovine resistance, genetic selection

## Abstract

**Simple Summary:**

Theileriosis is a potentially fatal disease for susceptible cattle and is an important cause of economic losses for farms around the world. Nevertheless, there are animals that can be protected with vaccination or acaricides, or that can recover after treatment is carried out. However, the current trend is to try to reduce the use of drugs and to implement more sustainable strategies. The following review presents the parasite’s life cycle, the clinical signs and lesions resulting from infection by *Theileria* spp. and discusses current strategies to control the disease and the development of strategies based on the genetic selection of resistant animals.

**Abstract:**

Diseases caused by ticks have a high impact on the health, welfare, and productivity of livestock species. They are also an important cause of economic losses in farms worldwide. An example of such diseases is theileriosis, which can be controlled by drugs or vaccines, although these are not fully efficient. Therefore, there is a need to develop alternative and more sustainable and efficient complementary strategies. These may involve the identification and selection of animals more resistant to the disease. Several previous studies have identified significant differences in resistance between different breeds, with resistant breeds typically identified as those native to the region where they are being studied, and susceptible as those from exotic breeds. These studies have indicated that resistance traits are intrinsically related to the modulation of the immune response to infection. This review aims to systematize the general knowledge about theileriosis, emphasize resistance to this disease as a sustainable control strategy, and identify which traits of resistance to the disease are already known in cattle.

## 1. Introduction

Worldwide, various animal diseases have negative impact on performance and welfare, and cause mortality and morbidity—a limiting factor to the sustainability and profitability of livestock production and to carbon-neutral farming [1,2]. Furthermore, certain animal diseases pose a threat to human health due to zoonotic transmission. Although there are several ways to minimize their incidence, such as resorting to preventive medicines and vaccinations, there are pressures on farmers to reduce reliance of production systems on control strategies such as extensive antibiotic and chemical usage [2,3]. This is motivated by the need to obtain chemical and drug-free animal products, to avoid antibiotics, anthelmintics and acaricides resistance, and to minimize the impact on animal welfare [4,5,6,7]. Therefore, there is a need for complementary control strategies, such as the selection and breeding of animals with increased resistance to infection and disease [3,8].

Disease resistance results from genetic variation found in comparing animals of different breeds [9]. Several studies have reported greater resistance in cattle of indigenous breeds compared to exotic breeds. The analyzed characteristics that prove the different resistance between breeds are associated with the severity of the clinical signs manifested by the animals, the exuberance of the immune response to the infection, and the differential expression of genes identified as candidate genes differentially expressed between breeds [10].

In this review, we will provide an overview of the life cycle of *Theileria* spp., the clinical pathological profile presented by infected animals, the strategies for controlling Theileriosis, and emphasize on the genetic selection of animals more resistant to the disease.

## 2. Bovine Theileriosis

### 2.1. Bovine Theileriosis—Definition

*Theileria* spp. are hemoparasites belonging to the Phylum Apicomplexa [11,12]. The species of this intracellular protozoa that infects bovines include *T. annulata*, *T. parva*, *T. mutans*, *T. orientalis complex*, *T. tarurotragi*, *T. velifera*, *T. sinensis*, and *Theileria* spp. *Yokoyama* [13,14]. *Theileria* spp. are transmitted by ixodid ticks of the genera *Amblyomma*, *Haemaphysalis*, *Hyalomma*, and *Rhipicephalus*, and the species is determined by their geographical location [14,15]. Although there are many *Theileria* species, only a few, particularly *Theileria parva* and *Theileria annulata*, are associated with severe clinical disease in cattle. *Theileria parva* occurs in eastern and southern Africa and is transmitted by *Rhipicephalus appendiculatus* ticks. *Theileria annulata* is widespread across the Mediterranean basin, northeast Africa, the Middle East, India, and Southern Asia, and is transmitted by several species of *Hyalomma* ticks [12].

### 2.2. Prevalence

In Table 1, we can see that the prevalence of *Theileria annulata*, responsible for Tropical Theileriosis, varies from less than 10% (Greece, Turkey, Pakistan, Ethiopia, and South Sudan) to values greater than or equal to 50% (Bangladesh, Sudan, Egypt, and Tunisia). The existence of high prevalence values and wide distribution may be associated with the use of the Holstein cattle breed for its excellence in milk production, increased animal movement, and climate change [16].

### 2.3. Economic Losses from Theileriosis

Tick-borne diseases are the cause of serious economic losses to livestock farming. There are recent studies that indicate an increase in the spread of ticks and tick-borne diseases, due to climate and environmental changes, which affect domestic ruminants and humans, leading to annual losses of USD 13.9 to 18.7 billion [34]. One of the critical tick-borne diseases of domestic cattle is theileriosis, which is caused by several *Theileria* species in tropical and subtropical countries. Some of these species causes disease outbreaks, high rates of mortality and morbidity, decreased production, and, consequently, serious economic losses [13,15,35,36]. According to Perera et al. (2014), dairy cows suffering from severe oriental theileriosis (resulting from infection by *Theileria orientalis*) produce 624 L less per capita at 305 days of lactation, leading to an estimated annual economic loss of AUD 202 (Australian dollars) (the equivalent of USD 179). In addition to the losses associated with production, there are also losses from mortality. In 2003 in the state of Victoria, Australia, the mortality rate from eastern theileriosis was 11%. The annual per capita cost (approximately AUD 227) associated with eastern theileriosis is significant and appears to be comparable to that of *T. parva* infection in Tanzania. Notably, the Tanzanian study included not only the costs associated with reduced milk production but also the costs of weight loss, acaricide treatments, and immunization [37]. It is estimated that worldwide there are around 250 million cattle at risk of tropical theileriosis (resulting from infection by *Theileria annulata*) [38]. In a study carried out in Pakistan, losses of 13.83% of farm income were estimated to be due to tropical theileriosis [23]. Considering the economic impact of theileriosis on livestock production, investment in research into sustainable control strategies for the disease is imperative to reduce losses in livestock production and ensure food and nutritional security worldwide.

### 2.4. Pathogenesis

In susceptible animals, pathogenic *Theileria* species cause acute lymphoproliferative diseases, with high levels of morbidity and mortality. This parasite infects nucleated cells such as monocytes, macrophages, T cells (CD4+ e CD8+), and B cells (B1 and B2), but also erythrocytes. The type of nucleated blood cells targeted by the parasite differ according to the species of *Theileria*. Nonetheless, pathogenicity is attributable to the life cycle stage in nucleated cells [12,39,40,41]. The aim of the parasite is to ensure its survival and increase its population, while increasing the probability of its transmission [40].

### 2.5. Life Cycle of Theileria spp.

The life cycle of *Theileria* spp. occurs among its invertebrate and vertebrate hosts. In vertebrate hosts, *Theileria* spp. infects ruminants and equids; in invertebrate hosts *Theileria* spp. infects ticks [40]. Initially, the host infection occurs with the invasion of sporozoites, transmitted through saliva secretion when an infected tick takes a meal. The sporozoite invasion occurs rapidly after entry, and the sporozoite nuclei divide to form a schizont. The schizont presents 30 nuclei in approximately 20 h as it approaches the nucleus of the host cell [40,42]. When leukocyte division occurs, the schizont is also divided because it is tightly bound to mitotic spindles [40,43]. Host cell transformation and proliferation is induced by the parasite. Depending on the *Theileria* species there may be multiple divisions in the lymphocyte stage and few or no divisions at the erythrocyte stage (e.g., *Theileria parva*), but in other species, there is little or no intralymphocytic multiplication, and *Theileria* multiplies essentially in the erythrocyte stage (e.g., *Theileria mutans*) [42].

A proportion of schizonts differentiates into merozoites that invade erythrocytes. When the multiplication is in the lymphocyte stage, at the end of multiplication, the schizont initiates transition into the next development stage, i.e., merozoites, which then infect erythrocytes. Merozoites are also known as piroplasms due to the fever they provoke [40]. Infected erythrocytes infect ticks when they take a meal, and gametogenesis and fertilization take place in the gut lumen of ticks. Thus, merozoites develop into male and female gametes that fuse, and sexual reproduction occurs, producing a zygote. The zygote or ookinete invades a gut epithelial cell where it remains during the tick molt cycle and develops into a single motile kinete that migrates to tick salivary glands [40,42]. In specific acini cells of the salivary gland, the parasite undergoes multiple divisions and generates thousands of sporozoites. The life cycle is completed only when mature sporozoites are transmitted from a tick to a new ruminant host in the later stages of feeding (Figure 1) [40,41]. Transmission in the tick is transstadial whereby larvae or nymphs can become infected [42].

### 2.6. Clinical Signs

The first clinical sign of theileriosis in cattle typically appears 7 to 15 days after attachment of the infected tick [45]. The most common sign is an increase in body temperature, which can reach 41.1 °C. In addition, the animal may have anorexia, pale mucous membranes (hemolytic anemia), jaundice, hemoglobinuria, swollen lymph lumps, loss of body condition, presence of petechiae on the conjunctiva, and the presence of ticks on the animal’s body [45,46,47]. Other clinical signs include lethargy, depression, corneal opacity, tachycardia, tachypnea, dyspnea, nasal discharge, diarrhea, reduced production, stillbirths, and miscarriages. In the final phase of clinical evolution, in serious cases, before death, the animal is typically in lateral recumbency, with hypothermia and severe dyspnea due to pulmonary edema [45,47]. The presence of these clinical signs, as well as their intensity, may differ according to the species and genotype of *Theileria* spp. that infected the animal [45,46,47].

### 2.7. Diagnosis

The diagnosis of theileriosis may be based on the use of traditional or molecular diagnostic methods. Traditional diagnostic methods include the identification of the aforementioned clinical signs present in infected animals, detection of findings in postmortem evaluations, and microscopic and serological evaluation. In the postmortem evaluation, pathological changes such as jaundice, pallor and enlargement of the liver, kidney, and spleen, hemorrhagic duodenitis, ulcers in the abomasal mucosa, pulmonary edema, and enteritis may be detected [47]. The microscopic evaluation includes identification of the parasite in red blood cells in blood smears stained with Giemsa. This method can be also used to estimate the degree of parasitemia, but it is only possible when the number of infected erythrocytes is high. Finally, the Immunofluorescence Antibody Test (IFAT: Sensitivity (Se)—71%; Specificity (Sp)—93% [48]), enzyme immunoassay (ELISA: Se—93.5%; Sp—93.5% [49]), and latex agglutination test can be performed [13,47]. These are serological methods which have a higher sensitivity compared to traditional methods. The specificity is relatively good, but generally lower than that of molecular methods. Molecular methods can overcome this limitation, in particular polymerase chain reaction (PCR: Se—83%; Sp—93% [48]), reverse line transfer hybridization assay (RLB), loop-mediated isothermal amplification (LAMP), real-time/quantitative PCR (qPCR: Se—97.1%; Sp—97.4% [50]) using hydrolysis probes, and multiplexed tandem PCR (MT-PCR: Se—98%; Sp—98.9% [50]) assays. Thus, molecular methods have greater specificity and sensitivity than traditional and serological methods, allowing the detection, characterization, differentiation, and quantification of different species of *Theileria* spp. [47].

## 3. Theileriosis Control Strategies

Theileriosis control strategies includes measures the pathogen (acaricides), cattle (e.g., vaccination, culling diseased animals and selection of resistant animals), or environmental control (e.g., biosecurity, sanitation, etc.) [51]. At present, there are no effective vaccines or drugs to control bovine theileriosis. The use of acaricides to eliminate the vector and reduction of cattle movement from nonendemic to endemic areas are the main methods of disease control [52]. As a preventive measure, cattle are routinely treated with synthetic pyrethroids prior to being put out to graze in pastures [53]. However, the widespread use of acaricides has resulted in an increased tick resistance to these chemical compounds. Acaricide resistance results from the selection of specific hereditary traits in a tick population due to exposure of the population to an acaricide, which results in an increase in the number of ticks that will survive after administration of the recommended dose of the acaricide in question. The main mechanisms that prevent the action of chemicals in ticks and that are responsible for this resistance are increased metabolic detoxification and point mutations at target sites. This resistance has a strong negative impact on tick and tick-borne disease control in cattle [54]. Cumulatively, the continued use of acaricides becomes economically unsustainable [51,55].

Buparvaquone, a known antiprotozoal, is used in the treatment of theileriosis, although it is not approved for the treatment of livestock in many countries, including European countries since it is not approved by the EMA (European Medicines Agency) [52,53,56,57,58,59]. Other substances are commonly used to treat theileriosis, such as oxytetracycline, imidocarb, halofunginone, or erythromycin, but to little effect [52,53,56]. According to the variation in clinical manifestations, additional drugs are used [57,58]. For example, in animals with a high temperature, Meloxicam or Paracetamol are indicated. In animals with lameness and difficulty in getting up, sodium acid phosphate is useful [57]. A symptomatic treatment option for extremely anemic animals is blood transfusion. In these animals, a solution with hydroxy ethyl starch in isotonic sodium chloride intravenous infusion is administered [52,57].

In recent times, there has been investment in the development of vaccines, although their realized ability to protect against certain *Theileria* species, when animals are naturally infected, is not yet fully known [52]. Some of these vaccines are produced via the culture of cells infected with the attenuated *T. annulata* schizont [55,56,58]. On the other hand, development of subunit vaccines is generally regarded as problematic for apicomplexan parasites due their genetic diversity. While there are promising initial results, further investment in research and the development of effective and affordable vaccines will be needed [52,53].

## 4. Genetic Selection—Resistance and Tolerance

Traditional genetic evaluations for animal breeding are based on statistical analysis to determine the genetic merit of an animal (the animal’s value as a parent, breeding value [60]) using pedigree information and performance records. The use of these methods, based on the phenotypic performance of the animals, led to noticeable improvements in many characters of economic importance in livestock species. However, these methods proved to be limiting or inefficient when the characters are difficult to measure or have a low heritability, as well as for those traits that should be evaluated in a large number of animals or are expressed very late, sometimes after the animal’s death. When the selection objective is the improvement of several traits (e.g., milk production and milk protein), with unfavorable correlation, the selection also does not become very efficient [61].

With advances in molecular genetics, the possibility of identifying DNA polymorphisms dispersed throughout the genome of various animal species, associated with productive traits of interest, has opened up, allowing the implementation of selection programs based on molecular markers or selection assisted by molecular markers (MAS) [62]. MAS does not replace the traditional selection techniques of quantitative genetics, but reinforces them, being particularly beneficial for traits with low heritability, with measurement difficulties or expense which cannot be evaluated during the animal’s life [63,64].

Technological advances in recent years have made it possible to generate molecular data, representing a true revolution in mass identification and genotyping of markers. High-density DNA chips were created to genotype tens of thousands to hundreds of thousands of markers in a single analysis [62,64]. These new technologies allowed the generation of new applications, such as methodologies for genetic evaluation and selection based on genomic value (Genomic Estimated Breeding Value—GEBV), with direct use in animal production worldwide. The genotypic information provided by DNA testing should help to improve the accuracy of selection and increase the rate of genetic progress by identifying animals carrying desirable genetic variants for a given trait at an earlier age [64].

Host populations will carry a wide variety of polymorphic genes, some of which will be associated with the response variation between individuals when they are infected by different pathogens [65]. Selection for disease resistance is much more difficult than selection for production traits because these can be measured directly or indirectly on each animal. Moreover, selection for disease resistance is easier when we only have one disease under study [2]. However, it is possible that interactions between the animal’s genotype and environment, which, if significant, the animals selected as more resistant in one environment may not be in another [66].

It will be interesting to invest in the development of alternative or complementary strategies, which may include breeding for increasing host resistance to infection or disease. Host genetic variability in disease resistance is due to variability in host immune responses to infection. Thus, it will be possible to improve the genetic resistance of animals to some diseases, although phenotypic determination of resistance in the field will be challenging. This could be costly and logistically difficult, making it a limitation to selection for disease resistance. For this reason, the genomic approach will be extremely advantageous, selecting animals based on their DNA. This can be achieved using major genes, QTL (quantitative trait loci) for resistance or SNP-chip (single-nucleotide polymorphism—chip)-based genomic predictors [3,65].

First, it is important to distinguish between the terms “infection”, which refers to invasion by a pathogen or parasite, and “disease”, which refers to the negative consequences of the host being infected, by the manifestation of clinical signs [3].

Disease resistance implies that the host has a negative impact on the fitness of the pathogen, causing its death [67]. Thus, resistance describes the host’s ability to limit pathogen load [68]. The mechanisms involved in resistance depend on the biological and immune response of the host, but also on the pathogens [2,67]. Animals that are not infected are the most resistant to infection and are the most useful to the producer. Therefore, genomic studies should focus on resistance rather than tolerance [3]. To select resistant animals, it is necessary to identify phenotypes accurately and to have consistent genetic markers with high predictive values for a disease phenotype [2].

Disease tolerance is related to the impact of a given level of infection on the animal’s performance, namely, the reduction in performance in the presence of a pathogen load [3,69]. Disease tolerance is different from resistance because it promotes host health while having a neutral to positive effect on pathogen fitness [67]. However, tolerance only manifests itself in infected animals, so the usefulness of selecting for this trait only arises when the prevalence of the disease is high. Resilience, a closely related concept, concerns the productivity of an animal in the face of infection. Resilience becomes a useful concept when all animals are infected [69].

Although genomic selection focuses on resistance, it has the disadvantage of, in some circumstances, imposing selective pressure on the pathogen. At this point, tolerance presents itself as a stable evolutionary strategy, which does not necessarily exert selective pressure on the pathogen. On the other hand, the selection of disease-tolerant animals presents itself as a negative strategy when there are unselected animals in the same environment [9].

## 5. Immune Response Mechanisms

The immune response, as a host defense and resistance mechanism, may differ according to the life cycle stage of the parasite [70]. In a first infection, the schizont phase is controlled by the innate immune response (natural killer cells (NK cells) and macrophages), but also by the acquired cellular immune response (cytotoxic T cells and T-helper cells) [71]. In turn, T-helper cells produce interleukin-2 (IL-2), necessary for the clonal expansion of cytotoxic T-lymphocytes, and interferon-γ (INF-γ), which activates macrophages to produce nitric oxide (NO). The latter can destroy the schizonts within the infected cells. Cytotoxic T-lymphocytes kill the infected target cells and cytokines (IL-2 and IFN-gamma) induce the production of specific antibodies. The function of natural killer cells is probably associated with lysis of infected cells or activation of macrophages through IFN-γ production. During infection, both B- and T-lymphocytes are activated [70]. On the other hand, high levels of antibodies have also been detected, although at a later stage, when the infection is already controlled [72].

## 6. Theileriosis Resistance

To identify genes that control a disease, and to develop new selection technologies, it is first necessary to identify animals that differ in their response to the disease, that is, in the expression of its phenotype. Thus, it will be useful to compare animals of different breeds which, during their evolution, have been subjected to different levels of infection pressure by different agents [73]. In the case of cattle, *B. taurus* and *B. indicus* belong to the same species but have great genetic variability [74]. Differences in the response to *T. annulata* infection were identified in a *B. indicus* breed (Sahiwal) and in a *B. taurus* breed (Holstein) [73,75]. Sahiwal is a breed indigenous to Pakistan recognized for its excellent resistance to internal and external parasites and its great capacity for subsistence milk production [76]. On the other hand, Holstein is the dairy breed with the highest productive capacity in the world. These animals are distinguished by their exceptionally high milk production capacity, the specific qualities of their udder and their ability to adapt to different soils and climates [77].

One of the parasites to which the Sahiwal breed is resistant are ticks, namely *Boophilus microplus*, the host of *T. parva* [73]. This decreases the ability of ticks to transmit *Theileria* sp., decreasing the incidence and severity of theileriosis, as well as the transmission of the parasite to other ticks [78]. Other *Bos indicus* breeds showed resistance to *T. annulata* and *T. sergentii*, as well as to the *T. annulata* host tick, namely *Hyalomma* spp. Despite this, the resistance mechanisms and underlying chromosomal regions or genes are generally not identified. There are limitations regarding studies that allow this identification to be carried out, as it is believed that host resistance to pathogens is polygenic. Therefore, it is possible to invest in the detection of candidate genes for resistance to theileriosis by identifying differences in gene expression in susceptible and resistant cattle at various key stages of infection [73]. There are studies that prove the differences in susceptibility to theileriosis in Holstein and Sahiwal cattle. In 2005, Glass et al. experimentally infected calves of both breeds with *T. annulata* sporozoites, verifying that Sahiwal calves survived without treatment, with lower body temperatures and with less parasitemia than Holstein calves, all of which showed severe responses. Furthermore, Sawihal calves showed a smaller increase in acute phase proteins alpha1 acid glycoprotein and haptoglobin than Holstein calves. Acute phase proteins are produced in response to the release of proinflammatory cytokines, which are believed to be responsible for the pyrexia, cachectic, and anorexic responses typical of theileriosis [75]. While a rapid and robust response of these proinflammatory cytokines can effectively block and eliminate the infection, increasing the duration and intensity of this response could also cause excessive tissue damage [79]. An analysis of the results obtained in this study allowed to identify the typical traits associated with resistance and to determine that the resistance mechanism is essentially associated with the schizont of the parasite, as it is at this time that the first clinical signs typically appear in the animal. In addition, it was possible to conclude that Sahiwal animals have an innate ability to limit the production of proinflammatory cytokines, and thus reduce immunopathology, as compared to Holstein animals [75].

In 2007, Glass et al. performed a global analysis of gene expression in macrophages from cattle of resistant (Sahiwal) and susceptible (Holstein) breeds. For this, they created a unique array focused on bovine macrophages, because they are the cells to which sporozoites have the greatest tropism [73,80]. They were able to determine that almost 600 genes were differentially expressed during the various phases of the life cycle of *T. annulata* in the cells of both breeds under study [73]. Microarray results showed that mRNA and CD14 (glycolipid-anchored membrane glycoprotein expressed in cells of the myelomonocyte lineage, such as macrophages) levels increased during the early stages of infection, with protein levels decreasing 15 days after infection [73,81]. Furthermore, the transcriptional response observed within 2 h differs from that presented after 72 h, which suggests the existence of temporal waves of gene transcription. The analysis of the microarray data allowed the identification of sixty-six transcripts that show specific differential expression according to breed during *T. annulata* infection. Patterns of gene expression differences may provide clues to underlying genetic differences, which may allow for the identification of candidate genes with causal polymorphisms [73].

Later, in 2008, Nafizi et al. evaluated the resistance to *T. annulata* of indigenous Iranian cattle compared to Holstein cattle, assuming that indigenous cattle are more resistant because they have a lower mortality rate from theileriosis. For this, they evaluated changes in the concentration of acute-phase proteins, namely Haptoglobin, Serum Amyloid A, ceruloplasmin, and fibrinogen. Thus, they found that Iranian indigenous cattle had lower parasitemia rate, milder clinical signs, and lower levels of all acute-phase proteins under study [82]. Kumar et al. (2016) used cattle from an indigenous breed of Pakistan (Tharparkar) and crossbred cattle in a study to determine gene expression in peripheral blood mononuclear cells infected in vitro with *T. annulata*. This study was based on the hypothesis that the susceptibility to tropical theileriosis shown by these breeds is due to dealing of infected cells with other immune cells, which will influence the immune response against *T. annulata*. This study, where differentially expressed genes were detected, confirmed the involvement in several pathways such as immune regulation, cell proliferation, cytoskeletal changes, and apoptosis [80]. Thus, several genes involved in *T. annulata* infection were detected with expression differences in the two classes of animals and a strong correlation with this infection [10,80]. For example, the invasive potential of transformed leukocytes after *Theileria annulata* infection involves TGF-b2 signaling [80]. It had already been detected that this gene is more expressed in Holstein Friesian cattle than in Sahiwal cattle, which indicated its correlation with susceptibility to the disease [83] In this work, the gene was negatively regulated in Tharparkar cattle, while its expression did not change in crossbred cattle, which may be associated with the lower susceptibility of indigenous cattle from Pakistan [80]. The TGFα gene, which regulates cell proliferation and migration, also showed upregulation in crossbred cattle and downregulation in Tharparkar cattle, which gives indication of the lower susceptibility of indigenous cattle. Other differentially expressed genes such as FCGR1A, SLC11A1, IGHG1, PRSS2, KLK12, and PDLIM1 showed significant differences in expression magnitude, this being crucial for the establishment of the parasite in the host, so they are essential regulators in the early stage of infection. Thus, the ability of *T. annulata* to affect immune processes, cell proliferation, apoptosis, and cytoskeletal organization is related to its ability to induce the expression of a key set of host genes that, in turn, lead to the activation/repression of pathways that are essential for the survival and growth of infected cells. The differences found between breeds will be a result of their individual ability to modulate host gene expression in response to infection [80].

## 7. Conclusions

Diseases transmitted by ticks, including theileriosis, cause serious economic losses to animal production. With all the limitations associated with the current forms of treatment and control of this disease, the ideal future approach would involve integrated strategies that are economically and environmentally sustainable. Selective breeding of cattle more resistant to theileriosis could offer a complementary and sustainable disease control option. There are already promising studies in this area, which have identified breeds with different susceptibilities to thick borne diseases, namely Sahiwal and Holstein cattle. This was confirmed by differences in the inflammatory response to infection and the identification of some genes that limit the response. Hence, further studies are needed to estimate the genetic variation and heritability for the resistance to theileriosis.

## Figures and Tables

**Figure 1 animals-12-02903-f001:**
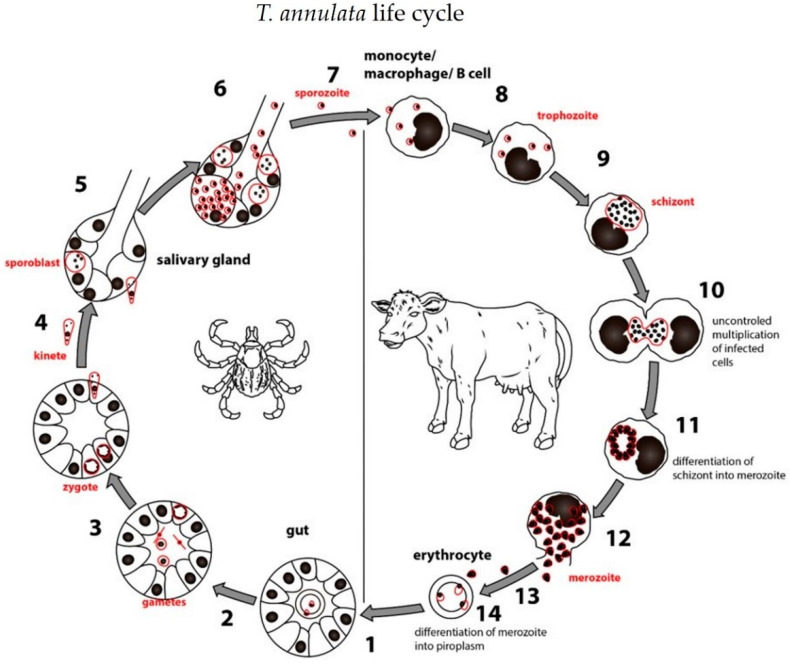
*T. annulata* life cycle as an example of the general life cycle of *Theileria* spp. Source: Andrade, et al [44] (Reprinted/adapted with permission from Ref. [44]. 2019, Pedro Andrade).

**Table 1 animals-12-02903-t001:** Prevalence of *Theileria annulata* in different countries in Southern Europe, Asia, and North Africa (PCR—Polymerase Chain Reaction, RLB—Reverse Line Blotting, IFA—Indirect Fluorescence Antibody, FRET-PCR— Fluorescence Resonance Energy Transfer Polymerase Chain Reaction).

Prevalence of *Theileria annulata*				
Country	Region	Technique	Prevalence	Reference
**Southern Europe**				
**Portugal**	All country	RLB	17.8%	[17]
**Spain**	Madrid	PCR	22.4%	[18]
**Italy**	Sicilia	IFA	26.0%	[19]
**Greece**	Macedonia	IFA	2.0%	[20]
**Asia**				
**Turkey**	All country	PCR	6.6%	[21]
**Iran**	Kerman	PCR	45.3%	[22]
**Pakistan**	Punjab	PCR	8.0%	[23]
**India**	Andhra Pradesh	PCR	32.4%	[24]
**Bangladesh**	Natore District	ELISA	80.0%	[25]
	Rajshahi District	ELISA	20.4%	[25]
**China**	Xinjiang Uygur Autonomous region	PCR	18.2%	[26]
**Northern Africa**				
**Ethiopia**	Humera	PCR	2.0%	[27]
**South Sudan**	Juba	RLB	0.2%	[28]
**Sudan**	Sennar State	PCR	50.0%	[29]
**Egypt**	Egyptian Oases	PCR	63.6%	[30]
**Tunisia**	Ariana	IFA	92.9%	[31]
**Algeria**	Central Algeria	FRET-PCR	30.0%	[32]
**Morocco**	Northwest of Morocco	PCR and DNA Sequencing	15.9%	[33]

## Data Availability

Not applicable.

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
