# Peer review of "Genetic Resistance of Bovines to Theileriosis"

_animals, 2022, doi:10.3390/ani12212903_

Round 1

Reviewer 1 Report (Previous Reviewer 1)

review complete

Author Response

Q. Review complete

A. Thank you for your positive comments.

Reviewer 2 Report (New Reviewer)

Authors reviewed the feneral knowledge of the Bovine Theileriosis, and emphasized the imnportance of the genetic resitance of Bovines to Theileriosis. Overall the topic is interesting, there are some major concerns:

1. The abstract is uninformative, authors should point out the background, and the objective of the current review at least;

2. The introduction section, more information abouth the gentic resistance of  of Bovines to Theileriosis should be provided;

3. 2.2 prevalence, why Portugal was selected as the case, if the review was limited in Portugal, then this information should be added in the title'

4. As the title "genetic resistance", the section "7. Theileriosis Resistance" should added more detail information.

Author Response

Q. Authors reviewed the feneral knowledge of the Bovine Theileriosis, and emphasized the imnportance of the genetic resitance of Bovines to Theileriosis. Overall the topic is interesting, there are some major concerns:

A. Thank you for the constructive comments.

The manuscript has been revised taking into consideration the reviewer’s comments.

Q. 1.The abstract is uninformative, authors should point out the background, and the objective of the current review at least;

A. The abstract has been restructured and the purpose of this review has been introduced.

Q. 2. The introduction section, more information abouth the gentic resistance of Bovines to Theileriosis should be provided;

A. The introduction section has been completed.

Q. 3. 2.2 prevalence, why Portugal was selected as the case, if the review was limited in Portugal, then this information should be added in the title'

A. Thank you for your comment. We decided to broaden the review on prevalence and included different countries in Europe, Asia and Africa.

Q. 4. As the title "genetic resistance", the section "7. Theileriosis Resistance" should added more detail information.

A. Thank you for your comment. Section 7 has been revised.

Round 2

Reviewer 2 Report (New Reviewer)

Authors have made corresponding revisions, no more comments

This manuscript is a resubmission of an earlier submission. The following is a list of the peer review reports and author responses from that submission.

Round 1

Reviewer 1 Report

minor spelling and grammatical errors

in the resistance section, the definitions used for 'resistance' and 'tolerance' do not hold up - just ensure the correct terminology linking to definitions is referred to with the studies identified

overall a well put together review

Author Response

REVIEWER #1: 
Thank you for the constructive comments. 
The manuscript has been revised taking into consideration the reviewer’s comments.

Q. Minor spelling and grammatical errors. 
A. Thank you for your comment. Our article has been evaluated by a professional proofreader so we can correct spelling and grammatical errors. 

Q. In the resistance section, the definitions used for 'resistance' and 'tolerance' do not hold up - just ensure the correct terminology linking to definitions is referred to with the studies identified. 
A. Thank you for this comment. The references were corrected.

Q. Overall a well put together review. 
A. We thank Reviewer #1 for the positive comments.

Reviewer 2 Report

This is a review about an important disease in cattle, and I found the paper interesting to read. As a general comment, I found the overview of the transmission, clinical pathology and diagnosis, sufficiently thorough and scientifically solid. Please find below more detailed comments and suggestions that can help improve clarity in the text.

However, the presentation of genetic selection as a complementary disease control strategy requires major revision, as, in its current form, it ignores fundamental facts and principles of genomic selection, and is misleading to the readers. Several statements are incorrect.  

Specific comments:

Summary:

Line 14: end of sentence, is that worldwide? Please add geographical location you are referring to.

Line 18: add ‘and discusses…’ after ‘spp.,’

Abstract

Lines 24-25: change ‘animals resistant’ to ‘animals more resistant’. This is important because, genetic selection, especially for polygenic complex traits, is not expected to make animals 100 % resistant – the phenotype is not binary, exactly because these traits are polygenic.

Line 25: change ‘There are already’ to ’Several previous studies’ and delete ‘that’

Introduction:

Line 29: ‘reduce production’, add ‘and welfare’

Line 32: You state ‘…diseases pose a threat to human health’. Please make this more specific by adding to the end of the sentence ‘via…’ for example you might mean zoonotic transmission?

Line 35: delete ‘reduction of usage’

Line 38: change ‘to apply’ to ‘for’

Line 38: replace ‘reproduction’ with ‘breeding’

Line 39: rephrase after comma to ‘with increased resistance to infection and disease’

Line 43: add ‘more’ resistant (see comment above)

Section ‘2. Bovine Theileriosis’:

This section is simply reporting facts and is missing the interpretation and comments of the authors. Although the paper is a review, we would still expect to hear the expert opinion of the authors. I would recommend to add in the text a couple of statements providing an overall assessment of the main trends reported.

The first paragraph speaks entirely about economic losses, hence I would change the title to be more informative and specific to, for example, ‘Economic losses from Theileriosis’.

In the section lines 136 – 152, Sensitivity and Specificity information is missing for the diagnostic methods.

In section lines 160 -163 it is odd to omit any mentioning/ discussion on AMR. Please include.

Line 46: You state: ‘an increase in the spread’. Also give the main reasons why that is the case.

Line 55: change ‘others, such as’ to ‘there are also losses’

Line 59 ‘Delete ‘In this study, with data from Tanzania, not only the’. And replace by ‘Notably, the Tanzanian study included…’. Line 60 ‘change were considered but’ to ‘and’

Line 61: add space in ‘thatworldwide’

Line 64-65. Redundant and confusing. For better clarity, I would suggest to delete ‘while in a previous… estimated’

Lines 66 – 75: This is a long paragraph reporting numbers. I would suggest to summarize the main numbers in a table and remove from text.

I don’t think Figure 1 is really informative. Perhaps if you could include a similar map for the main species infecting cattle (would link to what you state in lines 86 -89).

Lines 79 -89: move paragraph to the beginning of the section.

Line 82: delete ‘essentially’. Line 83: replace ‘are’ with ‘is’

Line 93: add ‘ the type of nucleated’

Line 102: Change ‘This’ to ‘The’

Line 108:  New paragraph starting with ‘A proportion of…’

Line 110: delete ‘essentially’

Lines 128 – 135: Are all the clinical signs the same for all species?

Line 142: delete ‘Furthermore’

Line 143: replace ‘is’ with ‘can be’. Line 144: change ‘and’ to ‘but’

Line 148: Delete ‘Thus, as molecular… conventional’

Section ‘3. Theileriosis control strategies’:

Line 154, change ‘decisions affecting’ to ‘measures’, change ‘the animal’ to ‘cattle’

Line 156: change ‘approaches’ to ‘control’

Lne 159: delete 'Thus, as’

Line 161: change ‘indiscriminate’ to ‘widespread’

Line 161 – 162: I would replace with 'increase tick resistance to these chemical compounds’

Line 167: change ‘can be’ to ‘commonly’. Line 169: Add ‘For example,’ before ‘… animals with high…’

Line 171: Rephrase to ‘A symptomatic treatment option for extremely anaemic animals, …’

Line 173: remove ‘can be’

Line 175: change ‘actual’ to ‘realized’

Line 176: change ‘results from’ to ‘are produced via’

Line 177: Are those subunit?

Line 183: Delete ‘is invariably’

Line 181: Start new section here for Genetic resistance

The rest of the paper is quite problematic. Please find specific corrections and suggestions below:

The sections ‘Resistance and Tolerance’ and ‘Genetic Selection’ can be combined in one. In fact, discussion about whether we should select for resistance or for tolerance should come after explaining genetic selection as an option.

Also, move paragraph for resistance, before paragraph for tolerance.

Line 203: Change ‘on the other hand’ to ‘Disease resistance’

Lines 207 – 208: This is important. In fact, it is an ongoing discussion in the research community about whether selecting for resistance places selective pressure on the pathogen, forcing pathogen evolution to overcome host’s resistance, and therefore we should target tolerance instead. There is also the consideration of the nature of the disease and the control strategy plan e.g. at national level. For example whether we aim to eradicate a disease (hence, it would make more sense to target resistance), or we can allow the disease to be endemic (hence target tolerance)? It would be interesting to see in this review the Authors’ thoughts on the topic specifically for Theileriosis.  

The next paragraph is more about immune response mechanisms, hence I would recommend a separate section for that, containing lines ‘212 - 224’.

Statement in lines 211-212 is incorrect. Genomic selection as a methodology does not require knowledge of the underlying biological pathway or the exact genes involved. That is fundamental.

Line 209: delete ‘their’, change ‘phenotype’ to ‘phenotypes’.

Move lines 209 – 212 to a next section about genetic selection.

Lines 226 – 229: This is speculation. Has the heritability for any trait associated to Theileriosis previously estimated? Do we know if there is any genetic variance in the trait? The presence of genetic variance would enable us to test genomic selection. If this has not been previously estimated, the Authors ought to highlight this research gap and the need for future studies.

Lines 230 – 231: Why is that? Please explain and include explanation in the text.

Lines 231 – 232: This is incorrect. According to what you have described in the previous paragraph about immune response, wouldn’t selection target better immune response? Then why is the case that resistance to one pathogen might lead to indirect selection for susceptibility to another? A more suitable example of indirect selection negatively affecting another trait of interest is the example of selecting for increased milk production in cattle, having a negative genetic correlation with fertility.

Line 235: change ‘there are’ to ‘it is possible that’

Line 238: delete ‘traditionally’

Line 240: I would suggest to clarify here: it requires phenotype and genotype data, the latter can be either pedigree (i.e. records of ancestry) information, or SNP (single nucleotide polymorphism) genotypes. If using SNP data, this is known as genomic selection.

Lines 241 – 245:  You need to be more specific, what would be the limitations for defining phenotypes specifically for Theileriosis? What would you suggest could be a suitable phenotype to select for this trait?

Line 246: Delete ‘Later’

Lines 253: QTLs and quantitative, complex traits: each gene has small individual effect and a large number of genes and the environment are involved in the control of the trait. Please be clearer with definitions and include in the text.

Section 6. Theileriosis resistance:

Lines 262 – 263: firstly, we would need to define phenotype(s).

Lines 256 – 273: This is very good. All these studies are indicative of the existence of genetic variation. This can be moved to the opening paragraph, lines 226 – 230.  

Line 279: change ‘multigenic’ to ‘polygenic’ throughout 

Line 280: Detecting candidate genes is indeed one thing we can do using genotype data, e.g. SNP genotypes. For example, via Genome Wide Association Studies. Has there ever been done a GWAS for Theileriosis? However, please note that knowing candidate genes is NOT REQUIRED for performing genomic selection to improve resistance nevertheless.

Conclusions:

Lines 317 – 318: Rephrase sentence to: ‘Selective breeding of cattle more resistant to Theileriosis could offer a complementary and sustainable disease control option.’

Add a concluding statement, for example, ‘Hence, further studies are needed to estimate genetic variation and heritability for resistance to Theileriosis’.

Author Response

Response to Reviewer 2 Comments
Thank you for the constructive comments. 
The manuscript has been revised taking into consideration the reviewer’s comments.

Q. This is a review about an important disease in cattle, and I found the paper interesting to read. As a general comment, I found the overview of the transmission, clinical pathology and diagnosis, sufficiently thorough and scientifically solid. Please find below more detailed comments and suggestions that can help improve clarity in the text. 
However, the presentation of genetic selection as a complementary disease control strategy requires major revision, as, in its current form, it ignores fundamental facts and principles of genomic selection, and is misleading to the readers. Several statements are incorrect. 
A. We thank Reviewer #2 for the positive comments. 

Q. Line 14: end of sentence, is that worldwide? Please add geographical location you are referring to. 
A. Corrected accordingly. 

Q. Line 18: add ‘and discusses…’ after ‘spp.,’ 
A. Corrected. 

Q. Abstract
Lines 24-25: change ‘animals resistant’ to ‘animals more resistant’. This is important because, genetic selection, especially for polygenic complex traits, is not expected to make animals 100 % resistant – the phenotype is not binary, exactly because these traits are polygenic. 
A. Corrected. 

Q. Line 25: change ‘There are already’ to ’Several previous studies’ and delete ‘that’ 
A. Corrected. 

Q. Introduction:
Line 29: ‘reduce production’, add ‘and welfare’ 
A. Corrected. 

Q. Line 32: You state ‘…diseases pose a threat to human health’. Please make this more specific by adding to the end of the sentence ‘via…’ for example you might mean zoonotic transmission? 
A. Corrected. 

Q. Line 35: delete ‘reduction of usage’ 
A. Corrected. 

Q. Line 38: change ‘to apply’ to ‘for’ 
A. Corrected. 

Q. Line 38: replace ‘reproduction’ with ‘breeding’ 
A. Corrected. 

Q. Line 39: rephrase after comma to ‘with increased resistance to infection and disease’ 
A. Corrected. 

Q. Line 43: add ‘more’ resistant (see comment above) 
A. Corrected. 

Q. Section ‘2. Bovine Theileriosis’:
This section is simply reporting facts and is missing the interpretation and comments of the authors. Although the paper is a review, we would still expect to hear the expert opinion of the authors. I would recommend to add in the text a couple of statements providing an overall assessment of the main trends reported.
A. Thank you for your comment. The appropriate considerations have been made. 

Q. The first paragraph speaks entirely about economic losses, hence I would change the title to be more informative and specific to, for example, ‘Economic losses from Theileriosis’. 
A. Corrected accordingly. 

Q. In the section lines 136 – 152, Sensitivity and Specificity information is missing for the diagnostic methods. 
A. Corrected. 

Q. In section lines 160 -163 it is odd to omit any mentioning/ discussion on AMR. Please include. 
A. Corrected accordingly. 

Q. Line 46: You state: ‘an increase in the spread’. Also give the main reasons why that is the case. 
A. Corrected accordingly. The main reasons for the increased spread have been included. 

Q. Line 55: change ‘others, such as’ to ‘there are also losses’ 
A. Corrected. 

Q. Line 59 ‘Delete ‘In this study, with data from Tanzania, not only the’. And replace by ‘Notably, the Tanzanian study included…’. Line 60 ‘change were considered but’ to ‘and’ 
A. Corrected. 

Q. Line 61: add space in ‘thatworldwide’ 
A. Corrected. 

Q. Line 64-65. Redundant and confusing. For better clarity, I would suggest to delete ‘while in a previous… estimated’ 
A. Corrected. 

Q. Lines 66 – 75: This is a long paragraph reporting numbers. I would suggest to summarize the main numbers in a table and remove from text. 
I don’t think Figure 1 is really informative. Perhaps if you could include a similar map for the main species infecting cattle (would link to what you state in lines 86 -89). 
A. Thank you for your suggestion. However, we have chosen to reduce the text information and keep the map. 

Q. Lines 79 -89: move paragraph to the beginning of the section. 
A. Corrected accordingly. 

Q. Line 82: delete ‘essentially’. Line 83: replace ‘are’ with ‘is’ 
A. Corrected. 

Q. Line 93: add ‘ the type of nucleated’ 
A. Corrected. 

Q. Line 102: Change ‘This’ to ‘The’ 
A. Corrected. 

Q. Line 108: New paragraph starting with ‘A proportion of…’ 
A. Corrected. 

Q. Line 110: delete ‘essentially’ 
A. Corrected. 

Q. Lines 128 – 135: Are all the clinical signs the same for all species? 
A. No. The presence of these clinical signs, as well as their intensity, may differ according to the species and genotype of Theileria spp. that infected the animal. Corrected. 

Q. Line 142: delete ‘Furthermore’ 
A. Corrected. 

Q. Line 143: replace ‘is’ with ‘can be’. Line 144: change ‘and’ to ‘but’ 
A. Corrected. 

Q. Line 148: Delete ‘Thus, as molecular… conventional’ 
A. Corrected. 

Q. Section ‘3. Theileriosis control strategies’:
Line 154, change ‘decisions affecting’ to ‘measures’, change ‘the animal’ to ‘cattle’ 
A. Corrected. 

Q. Line 156: change ‘approaches’ to ‘control’ 
A. Corrected. 

Q. Line 159: delete 'Thus, as’ 
A. Corrected. 

Q. Line 161: change ‘indiscriminate’ to ‘widespread’ 
A. Corrected. 

Q. Line 161 – 162: I would replace with 'increase tick resistance to these chemical compounds’ 
A. Corrected. 

Q. Line 167: change ‘can be’ to ‘commonly’. Line 169: Add ‘For example,’ before ‘… animals with high…’ 
A. Corrected. 

Q. Line 171: Rephrase to ‘A symptomatic treatment option for extremely anaemic animals, …’ 
A. Corrected. 

Q. Line 173: remove ‘can be’ 
A. Corrected. 

Q. Line 175: change ‘actual’ to ‘realized’ 
A. Corrected. 

Q. Line 176: change ‘results from’ to ‘are produced via’ 
A. Corrected. 

Q. Line 177: Are those subunit? 
A. Corrected. 

Q. Line 183: Delete ‘is invariably’ 
A. Corrected. 

Q. Line 181: Start new section here for Genetic resistance 
A. Corrected accordingly. 

Q. The rest of the paper is quite problematic. Please find specific corrections and suggestions below: The sections ‘Resistance and Tolerance’ and ‘Genetic Selection’ can be combined in one. In fact, discussion about whether we should select for resistance or for tolerance should come after explaining genetic selection as an option. Also, move paragraph for resistance, before paragraph for tolerance. 
A. Corrected accordingly. 

Q. Line 203: Change ‘on the other hand’ to ‘Disease resistance’ 
A. Corrected. 

Q. Lines 207 – 208: This is important. In fact, it is an ongoing discussion in the research community about whether selecting for resistance places selective pressure on the pathogen, forcing pathogen evolution to overcome host’s resistance, and therefore we should target tolerance instead. There is also the consideration of the nature of the disease and the control strategy plan e.g. at national level. For example whether we aim to eradicate a disease (hence, it would make more sense to target resistance), or we can allow the disease to be endemic (hence target tolerance)? It would be interesting to see in this review the Authors’ thoughts on the topic specifically for Theileriosis. 
A. Thank you for your comments. This topic has been improved.

Q. The next paragraph is more about immune response mechanisms, hence I would recommend a separate section for that, containing lines ‘212 - 224’. 
A. Corrected accordingly. 

Q. Statement in lines 211-212 is incorrect. Genomic selection as a methodology does not require knowledge of the underlying biological pathway or the exact genes involved. That is fundamental. 
A. Thank you for your comment. This sentence has been deleted. 

Q. Line 209: delete ‘their’, change ‘phenotype’ to ‘phenotypes’. 
A. Corrected. 

Q. Move lines 209 – 212 to a next section about genetic selection. 
A. Corrected. 

Q. Lines 226 – 229: This is speculation. Has the heritability for any trait associated to Theileriosis previously estimated? Do we know if there is any genetic variance in the trait? The presence of genetic variance would enable us to test genomic selection. If this has not been previously estimated, the Authors ought to highlight this research gap and the need for future studies. 
A. This is the focus of our work. Thus, we are studying native Portuguese cattle breeds, for which this knowledge is still lacking. We hope to be able to reach these conclusions. 

Q. Lines 230 – 231: Why is that? Please explain and include explanation in the text. 
A. Explanation given in the text. 

Q. Lines 231 – 232: This is incorrect. According to what you have described in the previous paragraph about immune response, wouldn’t selection target better immune response? Then why is the case that resistance to one pathogen might lead to indirect selection for susceptibility to another? A more suitable example of indirect selection negatively affecting another trait of interest is the example of selecting for increased milk production in cattle, having a negative genetic correlation with fertility. 
A. Corrected accordingly. 

Q. Line 235: change ‘there are’ to ‘it is possible that’ 
A. Corrected. 

Q. Line 238: delete ‘traditionally’ 
A. Corrected. 

Q. Line 240: I would suggest to clarify here: it requires phenotype and genotype data, the latter can be either pedigree (i.e. records of ancestry) information, or SNP (single nucleotide polymorphism) genotypes. If using SNP data, this is known as genomic selection. 
A. Corrected.

Q. Lines 241 – 245: You need to be more specific, what would be the limitations for defining phenotypes specifically for Theileriosis? What would you suggest could be a suitable phenotype to select for this trait? 
A. The answer was included in the text. 

Q. Line 246: Delete ‘Later’ 
A. Corrected. 

Q. Lines 253: QTLs and quantitative, complex traits: each gene has small individual effect and a large number of genes and the environment are involved in the control of the trait. Please be clearer with definitions and include in the text. 
A. Thank you for this comment. Corrected accordingly. 

Q. Section 6. Theileriosis resistance:
Lines 262 – 263: firstly, we would need to define phenotype(s). 
A. Answered in the text before the indicated line. 

Q. Lines 256 – 273: This is very good. All these studies are indicative of the existence of genetic variation. This can be moved to the opening paragraph, lines 226 – 230. 
A. Thank you for the excellent suggestion. This change has been made.

Q. Line 279: change ‘multigenic’ to ‘polygenic’ throughout 
A. Corrected. 

Q. Line 280: Detecting candidate genes is indeed one thing we can do using genotype data, e.g. SNP genotypes. For example, via Genome Wide Association Studies. Has there ever been done a GWAS for Theileriosis? However, please note that knowing candidate genes is NOT REQUIRED for performing genomic selection to improve resistance nevertheless. 
A. Currently, the focus of our work is to identify SNPs and genotypes associated with higher or lower resistance to Theileriosis. 

Q. Conclusions:
Lines 317 – 318: Rephrase sentence to: ‘Selective breeding of cattle more resistant to Theileriosis could offer a complementary and sustainable disease control option.’ 
A. Corrected accordingly. 

Q. Add a concluding statement, for example, ‘Hence, further studies are needed to estimate genetic variation and heritability for resistance to Theileriosis’.
A. Corrected accordingly. Thank you for all your excellent suggestions.

Round 2

Reviewer 2 Report

You have made most of my suggested changes in the text and the paper overall reads much better now. However, the genetics section still lacks coherence. I would suggest some re-arranging of text to improve that – please see suggestions below:

Simple Summary: line 17: replace ‘addresses’ with ‘presents’

Introduction: first sentence, rephrase to ‘…animal diseases have negative impact on performance and welfare, and cause mortality and morbidity’. You don’t need to mention fertility separately, it is a performance trait.

Line 40: delete ‘farm’

Rearrange the following sections as follows – sub-headings would help the reader:

Start with ‘Bovine Theileriosis – Definition’ (lines 72 – 82)

Continue with ‘Prevalence’ (lines 83 – 92)

Then ‘Economic losses from Theileriosis (lines 47 – 70)

Also: line 49: add ‘due to’, and line 63: replace ‘and’ with ‘but’

Then describe ‘Pathogenesis’ etc. (lines 107 – 177)

Lines 167 -177: Good that you added the discussion about sensitivity and specificity, but you need to find and add some numbers from the literature, at least for the main diagnostic methods you mention.

Lines 188 – 194: Good that you added about acaricide resistance, but you need to mention the mechanisms –  ‘selection of specific hereditary traits… due to exposure’ – you need to explain this.

Genetics – this part of the paper remains problematic, unfortunately:

Lines 216 -218: Delete the first sentence.

Line 222: why/ how the previous sentence relates to the next? Delete ‘thus’

Line 222: speculation. It is a possibility. Mention a specific example where that happened. This possibility can be investigated and taken into account in a genetic selection programme.

Lines 223 -224: why? If it is genetic resistance related to genes controlling innate immune response, wouldn’t that cover a broad range of diseases?

Lines 231 – 236: These methodologies are widely implemented commercially nowadays in livestock breeding – the limitations that you mention exist, but can be managed and methodologies have been developed that allow us to do so. As written, it sounds like these limitations are prohibitive, which is (a) not true and (b) confuses the message of the paper that genetic selection is a possible alternative strategy for disease control. Please keep in mind that several diseases in a number of different farmed species have been commercially addressed via genetic approaches, perhaps some more reading of the literature is needed by the Authors.

Paragraph starting line 240… this paragraph belongs to the previous paragraph. You need to merge all this text into one coherent story. As it is, it doesn’t show a good understanding of genetic selection methods by the Authors.

Note – line 245: the marker could be within the gene.

Line 245: add ‘as a more evolutionary stable…’

Round 3

Reviewer 2 Report

This is the third version of this paper that I read. I would like to thank the Authors for their efforts to make the suggested corrections and improve their paper. Unfortunately the sections referring to genetics still contain numerous inaccurate statements, indicating poor understanding of the topic. Hence, I have recommended rejection of this paper in its current form. Perhaps a new version of this paper, where the genetics section will be much shorter, and under a different title perhaps indicating an overall review of bovine Theileriosis rather than focusing on genetics could be more appropriate for re-submission.